# Scalable Synthesis of TRPV1 Antagonist Bipyridinyl Benzimidazole Derivative via the Suzuki–Miyaura Reaction and Selective SeO_2_ Oxidation

**DOI:** 10.3390/molecules28020836

**Published:** 2023-01-13

**Authors:** Joon-Hwan Lee, Jiduck Kim, Hakwon Kim

**Affiliations:** 1Department of Applied Chemistry, Global Center for Pharmaceutical Ingredient Materials, Kyung Hee University, Yongin 17104, Republic of Korea; 2Daewoong Pharmaceuticals Co., Ltd., Yongin 17028, Republic of Korea

**Keywords:** TRPV1 antagonist, bipyridinyl benzimidazole, Suzuki–Miyaura reaction, selenium dioxide oxidation

## Abstract

In this study, a kilogram-scale synthesis of a potent TRPV1 antagonist, **1,** is described. To synthesize bipyridinyl benzimidazole derivative **1**, we have developed a scalable Suzuki–Miyaura reaction capable of providing a key intermediate, 6′-methyl-3-(trifluoromethyl)-2,3′-bipyridine **4**, on a kilogram scale. Then, unlike the existing oxidation reaction pathway, two synthetic routes that can be applied to mass production of bipyridinyl carboxylic acid intermediate **5** or aldehyde intermediate **6** were developed by appropriately controlling the oxidation reaction using a selenium dioxide oxidizing agent. Using our developed synthetic procedure, which includes Suzuki–Miyaura coupling, selective selenium dioxide oxidation, and benzimidazole formation, multi-kilogram-scale bi-pyridinyl benzimidazole derivative **1** can be synthesized.

## 1. Introduction

Drugs leading the neuropathic pain market include Neurontin (gabapentin) [1] and Lyrica (pregabalin) by Pfizer and Cymbalta (duloxetine) [2] by Eli Lilly, but these drugs are coanalgesics (analgesic adjuvant), not analgesics. Among coanalgesics, gabapentin, amitriptyline, and carbamazepine are drugs belonging to the anticonvulsant group. It is presumed that they show analgesic efficacy by lowering the excitability of the cell membrane and toning down the overactive nervous system, but the detailed mechanism is not yet known. In the case of pregabalin, an alkylated analog of GABA, it shows superior antiepileptic efficacy and efficacy against diabetic neuralgia and postherpetic neuralgia compared to gabapentin. However, the central nervous system side effects [3] of these drugs are emerging as a problem and the lack of drug efficacy is a disadvantage. Although these drugs show serious side effects and low efficacy, they are still being used because there are no specific treatments. Therefore, TRPV1 (transient receptor potential vanilloid 1) has emerged as a target protein in drug development for neuropathic pain, and interest in it has been continuously increasing. The vanilloid receptor (capsaicin receptor, transient receptor potential channel, vanilloid subfamily member 1; TRPV-1; vanilloid receptor-1; VR-1), the receptor for capsaicin ((6E)-N-[(4-Hydroxy-3-methoxyphenyl)methyl]-8-methylnon-6-enamide), has long been assumed to exist. Finally, it was cloned in 1997 and called vanilloid receptor subtype 1 (hereinafter referred to as “VR-1”) by Caterina et al. [4]. Located on small unmyelinated nerve fibers (C-fibers) and myelinated nerve fibers (A-fibers), VR-1 is known as an ion channel which plays an important role in sensitizing pain stimuli by introducing the strong influx of cations such as calcium and sodium ions into the nerve endings upon activation in response to external or internal stimuli. External stimuli capable of activating VR-1 are reported to include heat and acids as well as vanilloid compounds [5]. As internal stimuli to VR-1, there are leukotriene metabolites such as 12-hydroperoxyicosatetraenoic acid (12-HPETE) [6] and arachidonic acid derivatives such as anandamide [7]. On the basis of these physiological activities, VR1 has attracted attention as an integral controller playing a pivotal role in transferring various external injurable stimuli into nerve cells. According to a report, VR1 knock-out mice responded like normal mice to general stimuli but showed greatly reduced pain response to heat or thermal hyperalgesia, which reflects the importance of VR1 against noxious stimuli [8]. VR1 is concentratively expressed in primary sensory neurons [4], which are responsible for controlling the functions of internal organs such as the skin, bones, the bladder, the gastrointestinal tract, the lungs, and so on. In addition, it is distributed among other neurons in the central nervous system, the kidneys, the stomach, T-cells [9,10,11], and throughout the entire body. VR1 is inferred to play an important role in cell division and cellular signal control. Indications found, thus far, to be associated with the control mechanism of the activity of VR1 include pain, acute pain, chronic pain, neuropathic pain, postoperative pain, migraines, arthralgia, neuropathy, nerve injury, diabetic neuropathy, neurological illness, neurodermatitis, strokes, bladder hypersensitivity, irritable bowel syndrome, respiratory disorders such as asthma, chronic obstructive pulmonary disease, etc., irritation to the skin, eyes, and mucous membranes, itching, fever, gastric-duodenal ulcer, inflammatory intestinal diseases, urge incontinence, and an anti-obestic effect. Based on pharmaceutical mechanisms, both agonists and antagonists of VR1 may be used for the treatment of the abovementioned diseases. The pain alleviating effects of VR1 agonists show the pharmaceutical mechanism based on the desensitization of capsaicin-sensitive sensory nerves. That is, VR1 agonists cause pain and irritation of sensory nerves to desensitize them to other noxious stimuli. Due to the induction of pain in the early stage, VR1 agonists are developed only as local analgesics. In contrast, acting through the mechanism of blocking sensory nerves from recognizing pain signals, VR1 antagonists do not cause early pain or irritation and have been studied for use in the treatment of systemic diseases. Therefore, analgesics targeting TRPV1 have been suggested as ideal drugs that can effectively control pain without side effects. The mechanism of TRPV1 inhibition was presented by confirming its efficacy in various animal models of pain in combination with in vitro assays [12]. In our previous report, candidate compound **1,** with a novel parent nuclear structure in the form of bipyridinyl benzimidazole, was developed as a possible TRPV1 antagonist (efficacy confirmed in assay for inhibitory activity against vanilloid receptor, test of calcium influx of vanilloid receptor, assay for analgesic effect, and toxicity test) [13]. To further this research, it is necessary to synthesize candidate compound **1** on a kilogram scale (compared to laboratory-scale synthesis) for preclinical and clinical trials. However, since the developed laboratory-scale synthesis uses a microwave reactor and silica gel column chromatography purification, a new strategy is required to enable kilogram-scale synthesis. Hence, in this report, we describe the development of a scalable process for kilogram-scale preparation of TRPV1 antagonist compound **1**.

## 2. Result and Discussion

In the original method, target compound **1** was synthesized only up to the 10 g scale in three steps. This synthesis began with preparation of 6′-methyl-3-(trifluoromethyl)-2,3′-bipyridine **4** via the Suzuki–Miyaura reaction of 2-chloro-3-(trifluoromethyl)pyridine **2** with (6-methylpyridin-3-yl)boronic acid **3**. Compound **3** was prepared from 5-bromo-2-methylpyridine using a previous method [14,15]. The subsequent oxidation of the methyl group of intermediate **4** using potassium permanganate in a microwave reactor led to carboxylic acid **5**. To complete the synthesis, compound **5** was converted to the desired compound **1** by a two-step reaction including amine-carboxylic acid coupling with compound **7** followed by cyclization (Figure 1) [13].

However, there are several problems with scaling the original method, such as purification via silica gel column chromatography at all steps and use of reagents or solvents that are not suitable at large scale. In Step 1 of Figure 1, we selected tetrakis(triphenylphosphine) palladium as the Pd catalyst in the Suzuki–Miyaura reaction because it yields few impurities and is most economically favored, even at the kilogram scale, compared to other Pd catalysts such as Pd_2_(dba)_3_, PdCl_2_(PPh_3_)_2_, and PdCl_2_(COD). In addition, 1,2-dimethoxyethane, which is not suitable for large-scale synthesis, was exchanged for ethanol, with the same reactivity among nontoxic solvents. The desired 6′-methyl-3-(trifluoromethyl)-2,3′-bipyridine **4** was synthesized using a 2.2 M Na_2_CO_3_ solution as a base under reflux reaction condition for 4 h [16,17,18]. In this reaction, serious impurities, such as triphenylphosphine oxide (TPPO), were generated from tetrakis(triphenylphosphine) palladium. In order to remove these impurities, the extraction solvent was changed from ethyl acetate to hexane, which lowered the solubility of the impurities.

In Step 2, the oxidation reaction with a potassium permanganate in a microwave reactor is not feasible in a kilogram-scale synthesis. Thus, it was necessary to find an alternative for the oxidation to carboxylic acid reaction of the pyridinyl methyl group of intermediate **4** for the synthesis of 3-(trifluoromethyl)-[2,3′-bipyridine]-6′-carboxylic acid **5** (bipyridine carboxylic acid) that would be feasible for a scalable synthesis [Route 1]. The methods investigated as alternatives included using potassium permanganate/water [19], selenium dioxide/pyridine [20], and potassium hydroxide/oxygen/18-crown-6 ether [21]. When potassium permanganate oxidation is not carried out in a microwave reactor under high-temperature and high-pressure conditions, the reaction time is long and the yield is remarkably low, making the process unsuitable for mass production. The method using potassium hydroxide and 18-crown-6 ether in an oxygen atmosphere was excluded due to the relatively low yield and the disadvantage of using highly dangerous oxygen gas, which is a limit for multi-kilogram-scale production.

In the method using selenium dioxide, the reaction proceeded smoothly when refluxed in pyridine (Table 1). If the reaction temperature is low or the amount of selenium dioxide used is insufficient, the starting material may remain or the less oxidized bipyridine aldehyde **6** may be produced. In an oxidation using selenium dioxide, an excess of about three equivalents of selenium dioxide should be used to maximize the reaction yield. However, agglomerates were formed due to the excess selenium dioxide used in the reaction and oxidized selenium (black precipitates). To solve this problem, celite was added together with selenium dioxide during the reaction to prevent agglomeration, and black precipitates were also removed through celite filtration (Table 1). In order to effectively extract bipyridine carboxylic acid **5** from the aqueous solution layer after completion of the reaction, concentrated hydrochloric acid was added to adjust the acidity of the aqueous solution to about pH 2.0.

In Step 3, the formation of benzimidazole through amine-carboxylic acid coupling and subsequent cyclization generates many impurities, so it was necessary to optimize the reaction conditions and develop a crystallization method to replace chromatography purification. To optimize the conditions, we first explored the effects of temperature and reagent equivalents in reactions using 1,4-dioxane or THF as solvents. In both solvents, the reaction rate was significantly slowed at low temperatures; at room temperature, 1,4-dioxane showed better yield and cost-effectiveness than THF, so it was selected as the optimal solvent (Table 2). The greater the amount of reagent used is, the higher the yield is. With cost considerations, the optimal amount of reagents, such as 4-bromobenzene-1,2-diamine (**7**, BBD) and HBTU, was 1.1 equivalents. After the coupling reaction of Step 3 of Route 1, only the reaction solvent was removed without an extraction process. Subsequently, for cyclization, acetic acid and toluene in a ratio of 9 to 1 were added to the reactor, followed by stirring at 125 °C.

When the in situ cyclization reaction was complete, the solvent was removed, and the residue was first crystallized from acetonitrile without an extraction process to remove dimeric impurities and triphenylphosphine oxide (TPPO). The filtrate was redissolved in ethyl acetate and added to water. Then, the organic layer was separated and concentrated under reduced pressure, followed by secondary crystallization using acetonitrile. This optimized process allowed effective preparation of target compound **1** at the kilogram scale.

Since the synthesis including the currently developed Route 1 proceeds through amide coupling using expensive HBTU, a more efficient and practical synthesis method has been sought. It was found that oxidation with selenium dioxide in Step 2 of Route 1 produces a small amount of bipyridine aldehyde **6**. We thought that this compound **6** could be converted into benzimidazole compound **1**, as shown in Route 2 of Figure 2. Compound **4** was converted to a novel bipyridine aldehyde **6** using selenium dioxide in 1,4-dioxane under mild conditions [22]. We confirmed that benzimidazole compound **1** can be synthesized simply and economically from bipyridine aldehyde **6** through a one-pot reaction [23,24,25]. As a result, the benzimidazole compound **1** could be synthesized at a large scale by a one-pot reaction of compound **6** with BBD **7** in the presence of only 1,4-benzoquinone [26]. In order to suppress the generation of carboxylic acid by-products in the selective selenium dioxide oxidation reaction of compound **4**, various reaction conditions, including solvent and equivalents, were explored to minimize the oxidation to carboxylic acid reaction. Screening of various organic solvents in the oxidation reaction showed that 1,4-dioxane was the optimal solvent (Table 3).

The oxidation reaction with selenium dioxide was excellent under reflux conditions [27], but the oxidation to carboxylic acid reaction had the tendency to increase when the volume of the solvent was 10 volumes or less [28,29,30]. In addition, since the oxidation to carboxylic acid reaction greatly increases when more than three equivalents of selenium dioxide are used, the use of three equivalents of selenium dioxide was selected as the optimal condition (Table 4). The optimal conditions for the selective oxidation reaction using selenium dioxide were reaction of 3.0 eq selenium oxide at 110 °C in 1,4-dioxane of 10 volumes.

The reaction yield of Step 2 in Route 2 was about 60% at the 10 to 100 g scale but decreased to a yield of 40% at the multi-kilogram scale. When the oxidation reaction proceeds on a large scale, it seems that the production yield of the bipyridine aldehyde **6** is lowered because the oxidation to carboxylic acid reaction proceeds more quickly and produces more bipyridine carboxylic acid **5**.

According to reference [31], selenium dioxide is reduced to selenium (black precipitate) and H_2_O after the aldehyde formation reaction, and another SeO_2_ in the presence of pyridine derivatives is converted to H_2_SeO_3_ (selenous acid) by H_2_O [32]. Since bipyridine aldehyde **6** is oxidized to bipyridine carboxylic acid **5** by H_2_SeO_3_, it was considered that inhibition of H_2_SeO_3_ formation could prevent oxidation of the aldehyde. Although the pyridine used for carboxylic acid synthesis was not used in this reaction, bipyridine aldehyde **6** was obtained in low yield because the pyridine moiety of compound **4** would participate in the formation of selenous acid. Therefore, we attempted to suppress the oxidation to the carboxylic acid reaction by blocking the nitrogen moiety of pyridine by adding an acid. Screening results for acid showed that acetic acid was less reactive and sulfuric acid reacted slightly slower than nitric acid. Therefore, nitric acid was selected as the acid catalyst.

Compound **1** was synthesized in Step 3 of Route 2 with only one equivalent of BBD and 1,4-benzoquinone in acetonitrile at room temperature. After completion of the reaction, the solid residue was removed by passing the products through activated carbon, and the resultant was purified by crystallization from acetonitrile. This optimized process for Route 2 allowed kilogram-scale production with a simpler and more economical synthesis method than Route 1.

## 3. Materials and Methods

### 3.1. Chemicals and Instruments

The ^1^H NMR and ^13^C NMR spectra were recorded on a Bruker 500 MHz spectrometer (Germany) and Bruker 126 MHz spectrometer, respectively. Chemical shifts (*δ*) were reported in parts per million (ppm), and coupling constants (*J*) were reported in Hertz (Hz). The following abbreviations were used to describe multiplicities: s = singlet, d = doublet, t = triplet, q = quartet, m = multiplet, and br s = broad singlet. HRMS was recorded on a LTQ Orbitrap XL of Thermo Fisher Scientific instrument (Waltham, USA). All chemicals were purchased from Sigma-Aldrich (St. Louis, USA), Alfa Aesar (Haverhill, USA), Acros Organics (Brookline, USA), and Tokyo Chemical Industry (Tokyo, Japan) and used without further purification. Reaction progress was monitored by thin-layer chromatography (TLC, Merck kieselgel 60 F_254_), and column chromatography was performed using Merck silica gel 60 (230-400 mesh). Extra-pure-grade solvents for column chromatography were purchased through Samchun Chemicals (Seoul, Korea), Duksan Chemicals and Daejung Chemicals (Incheon, Korea). The 2-chloro-3-(trifluoromethyl)-pyridine (Du-Hope international group, China) and 6-methylpyridine-3-yl-3-boronic acid (Hanchem, Korea) were supplied through consignment production when used over kilograms.

### 3.2. Experimental Procedures

#### 3.2.1. Synthesis of 5-(3-(Trifluoromethyl)pyridin-2-yl)-2-methylpyridine (4)

2-Chloro-3-(trifluoromethyl)pyridine (25.0 kg, 137.7 mol), Pd(PPh_3_)_4_ (4.8 kg, 4.1 mol), and (6-methylpyridin-3-yl)boronic acid (22.8 kg, 166.6 mol) in ethanol (125.0 L) were stirred at 25 °C. After addition of 2.2 M sodium carbonate solution (29.2 kg in 125.0 L of water) at 25 °C, the mixture was heated to reflux for 4 h. After reaction completed, the mixture was cooled to ambient temperature. Ethanol was removed under reduced pressure. The aqueous NH_4_Cl solution (25.0 kg in 225.0 L of water) and hexane (200.0 L) were added and filtered through celite. The organic layers were separated and the aqueous layers was extracted with hexane (150.0 L), and the combined organic layers were dried over MgSO_4_. The organic layers were evaporated in vacuo at 40 °C, and additional hexane (75.0 L) was added and then heated to 40–50 °C. After complete dissolution, the mixture was cooled to 20–25 °C and stirred for 1 h. The slurry was filtered, and the filter cake was washed with hexane (25.0 L). The wet cake was dried in a vacuum oven at 25 °C for 12 h to give the compound **4** as a yellowish powder (30.5 kg) in 93.1% yield. ^1^H NMR (500 MHz, methanol-*d*) *δ* (ppm): 8.87 (dd, J = 4.9, 1.6 Hz, 1H), 8.51 (d, J = 2.3 Hz, 1H), 8.31 (dd, J = 8.1, 1.6 Hz, 1H), 7.86 (dd, J = 8.1, 2.4 Hz, 1H), 7.67 (dd, J = 8.1, 4.9 Hz, 1H), 7.44 (d, J = 8.0 Hz, 1H), 2.63 (s, 3H). ^13^C NMR (126 MHz, methanol-*d*) *δ* (ppm): 160.12, 156.09, 153.46, 148.98, 138.70, 136.75, 134.01, 126.61 (q, J = 31.92 Hz, CCF_3_), 124.97 (q, J = 273.13 Hz, CF_3_), 124.42, 124.24, 23.74. HRMS (ESI) (*m*/*z*): calcd for [C_12_H_9_F_3_N_2_]^+^ 239.07906 [M+H]^+^, found 239.07904. (Appendix A)

#### 3.2.2. Synthesis of 3-(Trifluoromethyl)-2,3‘-bipyridin-6′-carboxylic acid (5) by Route 1

5-(3-(Trifluoromethyl)pyridin-2-yl)-2-methylpyridine (**4**, 15.0 kg, 63.0 mol) and pyridine (60.0 L) were stirred at 25 °C. SeO_2_ (21.0 kg, 188.9 mol) and celite (4.5 kg) were added to a reactor, and the mixture heated to reflux for 4 h. The reaction mixture was cooled to ambient temperature and filtered under vacuum. The solid was washed with EtOAc (2 × 30.0 L), and the filtrate was concentrated under reduced pressure at 80–90 °C. EtOAc (120.0 L) and water (120.0 L) were added, and the solution was acidified to a pH in the range of 2.0 to 2.2 using c-HCl. The organic layers were separated, and the aqueous layers were extracted twice with EtOAc (2 × 120.0 L), and the combined organic layers were dried over MgSO_4_. The organic layers were evaporated in vacuo at 60 °C, and then additional ACN (15.0 L) was added followed by heating to 80 °C. After complete dissolution, the mixture was cooled to 25 °C and stirred for 1 h. The slurry was filtered, and the filter cake was washed with IPE (30.0 L). The wet cake was dried in a vacuum oven at 25 °C for 12 h to give the compound **5** as a brown powder (15.2 kg) in 90.3% yield. ^1^H NMR (500 MHz, methanol-*d*) *δ* (ppm): 8.92 (dd, J = 4.9, 1.6 Hz, 1H), 8.78 (d, J = 2.2 Hz, 1H), 8.34 (dd, J = 8.2, 1.6 Hz, 1H), 8.29 (d, J = 8.1 Hz, 1H), 8.14 (dd, J = 8.1, 2.2 Hz, 1H), 7.71 (dd, J = 8.1, 4.9 Hz, 1H). ^13^C NMR (126 MHz, methanol-*d*) *δ* (ppm): 167.12, 155.13, 153.66, 149.97, 149.34, 139.58, 139.33, 136.75, 126.67 (q, J = 32.17 Hz, CCF_3_), 125.94, 124.89, 124.86 (q, J = 273.21 Hz, CF_3_). HRMS (ESI) (*m*/*z*): calcd for [C_12_H_7_F_3_N_2_O_2_]^+^ 269.05324 [M+H]^+^, found 269.05322.

#### 3.2.3. Synthesis of 3-(Trifluoromethyl)-2,3‘-bipyridin-6′-carbaldehyde (6) by Route 2

5-(3-(Trifluoromethyl)pyridin-2-yl)-2-methylpyridine (15.0 kg, 63.0 mol) and 1,4-dioxane (150.0 L) were stirred at 25 °C. SeO_2_ (21.0 kg, 188.9 mol), HNO_3_ (1.0 L, 12.6 mol) and celite (3.8 kg) were added, and the mixture heated to reflux for 8 h. The reaction mixture was cooled to ambient temperature and filtered under vacuum. The solid was washed with EtOAc (120.0 L), and 7% NaHCO_3_ (180.0 L) was slowly added to organic layers. The combined organic layers were dried over MgSO_4_ and evaporated in vacuo to obtain the compound **6** as a brown sticky oil (11.1 kg) in 70.1% yield. ^1^H NMR (500 MHz, methanol-*d*) *δ* (ppm): 8.88 (dd, J = 5.0, 1.6 Hz, 1H), 8.62 (d, J = 2.1 Hz, 1H), 8.31 (dd, J = 8.1, 1.6 Hz, 1H), 7.99 (dd, J = 8.1, 2.2 Hz, 1H), 7.77 (d, J = 8.1 Hz, 1H), 7.67 (dd, J = 8.1, 4.9 Hz, 1H). ^13^C NMR (126 MHz, methanol-*d*) *δ* (ppm): 193.52, 160.99, 155.78, 153.47, 148.87, 138.85, 136.66, 136.21, 126.51 (q, J = 31.92 Hz, CCF_3_), 124.49, 124.88 (q, J = 273.25 Hz, CF_3_), 121.24. HRMS (ESI) (*m*/*z*): calcd for [C_12_H_7_F_3_N_2_O]^+^ 253.05832 [M+H]^+^, found 253.05835.

#### 3.2.4. Synthesis of 6-Bromo-2-(3-(trifluoro)-2,3′-bipyridin-6-yl)-1H-benzo[d]imidazole (1) by Route 1

3-(Trifluoromethyl)-2,3′-bipyridin-6′-carboxylic acid (**5**, 15.0 kg, 55.9 mol), 4-bromobenzene-1,2-diamine (11.5 kg, 61.5 mol) and HBTU (23.3 kg, 61.5 mol) in 1,4-dioxane (150.0 L) were stirred at 25 °C. The mixture was added dropwise to DIPEA (15.9 lg, 123.0 mol) and stirred at 25 °C for 5 h. The reaction mixture was evaporated in vacuo at 60 °C, and acetic acid (67.5 L) and toluene (7.5 L) were added to the concentrated residue. The mixture was stirred at 80 °C for 10 h, and then the reaction solution was evaporated in vacuo at 90 °C. ACN (120.0 L) was added to the concentrated residue, and it was stirred at 85 °C for 1 h. The reaction mixture was cooled to ambient temperature and stirred for additional 10 h. The slurry was filtered, and the filter cake was washed with ACN (30.0 L). EtOAc (150.0 L) and water (150.0 L) were added, and the solution mixtures were adjusted to a pH in the range of 7.0 to 7.5 using 4N NaOH aqueous solution with stirring. The organic layer was separated and the aqueous layers were extracted with EtOAc (75.0 L). The combined organic layers were dried over Na_2_SO_4_ (15.0 kg) and evaporated in vacuo at 60 °C. After adding ACN (60.0 L), the mixture was heated to 90 °C. After complete dissolution, the mixture was cooled to 25 °C and stirred for 1 h. The slurry was filtered, and the filter cake was washed with ACN (15.0 L). The wet cake was dried in a vacuum oven at 40 °C for 12 h to give the compound **1** as a white powder (15.7 kg) in 67.0% yield. ^1^H NMR (500 MHz, methanol-*d*) *δ* (ppm): 88.89 (dd, J = 4.8, 1.6 Hz, 1H), 8.81 (d, J = 2.1 Hz, 1H), 8.36 (d, J = 8.1 Hz, 1H), 8.31 (dd, J = 8.2, 1.6 Hz, 1H), 8.07 (dd, J = 8.1, 2.2 Hz, 1H), 7.79 (s, 1H), 7.67 (dd, J = 8.1, 4.9 Hz, 1H), 7.55 (d, J = 8.1 Hz, 1H), 7.39 (dd, J = 8.5, 1.9 Hz, 1H). ^13^C NMR (126 MHz, MeOD) δ (ppm): 153.84, 152.96, 151.32, 148.88, 148.17, 137.74, 135.62, 125.68, 124.01 (q, J = 32.09 Hz, CCF_3_), 123.68 (q, J = 273.92 Hz, CF_3_), 123.56, 120.92, 115.16. HRMS (ESI) (*m*/*z*): calcd for [C_18_H_10_BrF_3_N_4_]^+^ 419.01137 [M+H]^+^, found 419.01138. m.p.: 160-166 °C.

#### 3.2.5. Synthesis of 6-Bromo-2-(3-(trifluoro)-2,3′-bipyridin-6-yl)-1H-benzo[d]imidazole (1) by Route 2

3-(Trifluoromethyl)-2,3′-bipyridin-6′-carbaldehyde (6, 15.0 kg, 39.3 mol), ACN (30.0 L), 4-bromobenzene-1,2-diamine (7.3 kg, 39.3 mol) and 1,4-benzoquinone (4.2 kg, 39.3 mol) were added, and the mixture was heated to reflux for 2 h. The reaction solution was concentrated in vacuo at 50 °C, and EtOAc (75.0 L) was added. The activated carbon (1.5 kg) was added, followed by stirring for 30 min. It was filtered through celite and washed with EtOAc (45.0 L). The mixture was evaporated in vacuo at 50 °C, and after adding ACN (39.6 L), the mixture was heated to 90 °C. After complete dissolution, the mixture was cooled to 25 °C and stirred for 1 h. The slurry was filtered and the filter cake was washed with ACN (15.0 L). The wet cake was dried in a vacuum oven at 40 °C for 12 h to give the compound **1** as a white powder (11.9 kg) in 72.0% yield.

## 4. Conclusions

In conclusion, we developed a highly scalable synthesis of compound **1**, a potent TRPV1 antagonist. In this study, we developed a scalable method for synthesis of compound **1** in two different routes with high yield and quality that are capable of multi-kilogram-scale synthesis for non-clinical and clinical trials. In particular, it has been demonstrated that compound **1** can be successfully prepared at the kilogram scale using this process. Our remarkable improvements include scalable synthesis of intermediate **4** by optimized kilogram-scale Suzuki–Miyaura coupling and scalable synthesis of intermediate **5** or **6** by oxidation to carboxylic acid or oxidation via controlled selenium dioxide reactions. Finally, by developing a scalable synthetic method for bipyridinyl benzimidazole using intermediate **5** or **6**, synthesis of compound **1** in high yield is achievable at the multi-kilogram scale.

## Data Availability

Data are available within the manuscript and in the Appendix A.

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
