# Peer review of "Scalable Synthesis of TRPV1 Antagonist Bipyridinyl Benzimidazole Derivative via the Suzuki–Miyaura Reaction and Selective SeO2 Oxidation"

_molecules, 2023, doi:10.3390/molecules28020836_

Round 1

Reviewer 1 Report

The paper by Kim and collaborators deals with the scale-up synthesis of a drug for neuropathic pain. Although the target chemical is an important drug, the paper is just an optimization of a 3 steps synthetic route with not relevant improvements in the synthetic strategy (that remains the same) and thus not deserving publication on Molecules. The opinion of this referee is that the work can find publication in other more appropriate journals from MDPI (Compounds, Organics or Drugs and Drugs Candidates) after the following points will be addressed:

- the introduction part is poor, it needs to be improved for giving the reader a more complete vision of the problem the authors are trying to address with the paper

- accordingly, also the references should be improved. 10 references are really too few.

- the abstract is confusing, it should be rephrased to make clearer the paper topic

- several compounds are cited in the text with an incomplete name. If you want to report the compound name you should report the complete one (e.g. 6’-methyl pyridine and chloropyridine for compounds 4 and 2, respectively, are not correct)

- obviously, authors should remove Pd(PPh3)4 from the list of Pd compounds cheaper than tetrakis(tri-phenylphosphine) palladium

- the statement that “the use of oxygen gas is a challenge for kilogram-scale production” is not shareable. Molecular oxygen is one of the cleanest oxidant available

Author Response

Please, refer to our revised manuscript.

Reviewer 2 Report

The manuscript submitted by Kim and Co-workers deals with the large scale synthesis of biologically active Bipyridinyl Benzimidazole derivative. There is huge scope to modify the synthesized scaffold which may be taken care of by them afterward. It is basically a multistep process as the starting materials are very simple and easily available. Couple of well known process was used by them. Indeed, the work was meticulously done by them and this reviewer feels the manuscript may be accepted for publication in Molecules after addressing the followings.

·         Yield of each product in scheme 1 should be mentioned.

·         Synthesis of compound 4, has not been optimized and it is assumed some reported methods may be consulted by authors. It is essential to cite those reference papers.

·         Preparation of Compound 6 is not clearly described. Under which condition the aldehyde usually formed? Proper reference should be cited. Even though for the preparation of both compounds 5 & 6; the same quantity of SeO2 was used. What prompted the reaction to stop at aldehyde stage instead of acid? This part should be addressed carefully.

Author Response

(The authors gave the same response as above.)

Reviewer 3 Report

The manuscript submitted by Hakwon Kim et al. reports the synthesis, on a kilogram scale, of a TRPV1 antagonist via the Suzuki–Miyaura reaction. It is reasonably interesting simply because it gives the perspective of the industry of a multi-step synthesis and the difficulties of working on a kilogram scale (in my lab, we typically work on the gram/milligram scale!). So, I recommend the publication of the manuscript.

Some comments:

In the Introduction, there is no information about the methods of synthesis reported in the literature concerning the compounds discussed in the manuscript. A few references, particularly from the last 10 years, would be useful to readers of the paper.

In the title, and throughout the text, ‘Suzuki-Miyaura’ should be changed to Suzuki–Miyaura (with a en dash).

Page 1, line 4, ‘carbamazephine’ should be carbamazepine.

Page 1, the name of capsaicin (in parentheses) should be changed to the IUPAC name: (6E)-N-[(4-Hydroxy-3-methoxyphenyl)methyl]-8-methylnon-6-enamide

Page 2, line 2 – the name ‘12-hydroperoxyeicosatetraenoic acid’ should be corrected to the IUPAC name (it is ‘icosa’, not ‘eicosa’; locants for the double bonds are missing).

Page 2, last paragraph – The name ‘6-methylpyridin-3-yl-3-boronic acid’ should be corrected to (6-methylpyridin-3-yl)boronic acid. (see also the Materials and Methods section)

Page 2 and following – the expression ‘peptide coupling’ should be changed throughout the manuscript because it does not make sense in the context of this work (no amino acids or peptides are involved!).

Page 3 and following – ‘n-hexane’ is not an IUPAC name, it should be corrected to ‘hexane’.

Page 3 and following – The name of acid 5 should be corrected to ‘3-(trifluoromethyl)-[2,3'-bipyridine]-6'-carboxylic acid’. The contracted name ‘bipyridine carboxylic acid’ may be used.

Page 3 (last paragraph) and following – the contracted name ‘bipyridinyl aldehyde 6’ should be corrected to ‘bipyridine aldehyde 6’ or, preferably, to ‘bipyridine-carbaldehyde 6’.

Page 4 (Table 1) and following – the term ‘peroxidation’ is ambiguous and should be changed throughout the manuscript to ‘oxidation to carboxylic acid’. For instance, for table 1, is should be:

‘Table 1. Optimization of the oxidation reaction of compound 4 to carboxylic acid 5’.

Page 5 – where is “The oxidation reaction of selenium dioxide” should be “The oxidation reaction with selenium dioxide”

Page 5 – Please change “Table 4. Optimization of oxidation reaction conditions of compound 4” to “Table 4. Optimization of the reaction conditions for the oxidation of compound 4 to aldehyde 6”.

Author Response

Thank you for your comments.

We revised our manuscript in accordance with your advices.

Comments and Suggestions for Authors :

The manuscript submitted by Hakwon Kim et al. reports the synthesis, on a kilogram scale, of a TRPV1 antagonist via the Suzuki–Miyaura reaction. It is reasonably interesting simply because it gives the perspective of the industry of a multi-step synthesis and the difficulties of working on a kilogram scale (in my lab, we typically work on the gram/milligram scale!). So, I recommend the publication of the manuscript.

Some comments:

In the Introduction, there is no information about the methods of synthesis reported in the literature concerning the compounds discussed in the manuscript. A few references, particularly from the last 10 years, would be useful to readers of the paper.

  • There are no documents reported on the synthesis method of this compound, only patents. So, I explained it in detail in the introduction part.

In the title, and throughout the text, ‘Suzuki-Miyaura’ should be changed to Suzuki–Miyaura (with a en dash).

  • We agree with your opinion, and it was corrected.

Page 1, line 4, ‘carbamazephine’ should be carbamazepine.

  • We agree with your opinion, and it was corrected.

Page 1, the name of capsaicin (in parentheses) should be changed to the IUPAC name: (6E)-N-[(4-Hydroxy-3-methoxyphenyl)methyl]-8-methylnon-6-enamide.

  • We agree with your opinion, and it was corrected.

Page 2, line 2 – the name ‘12-hydroperoxyeicosatetraenoic acid’ should be corrected to the IUPAC name (it is ‘icosa’, not ‘eicosa’; locants for the double bonds are missing).

  • We agree with your opinion, and it was corrected.

Page 2, last paragraph – The name ‘6-methylpyridin-3-yl-3-boronic acid’ should be corrected to (6-methylpyridin-3-yl)boronic acid. (see also the Materials and Methods section)

  • We agree with your opinion, and it was corrected.

Page 2 and following – the expression ‘peptide coupling’ should be changed throughout the manuscript because it does not make sense in the context of this work (no amino acids or peptides are involved!).

  • We agree with your opinion, and it was corrected.

Page 3 and following – ‘n-hexane’ is not an IUPAC name, it should be corrected to ‘hexane’.

  • We agree with your opinion, and it was corrected.

Page 3 and following – The name of acid 5 should be corrected to ‘3-(trifluoromethyl)-[2,3'-bipyridine]-6'-carboxylic acid’. The contracted name ‘bipyridine carboxylic acid’ may be used.

  • We agree with your opinion, and it was corrected.

Page 3 (last paragraph) and following – the contracted name ‘bipyridinyl aldehyde 6’ should be corrected to ‘bipyridine aldehyde 6’ or, preferably, to ‘bipyridine-carbaldehyde 6’.

  • We agree with your opinion, and it was corrected.

Page 4 (Table 1) and following – the term ‘peroxidation’ is ambiguous and should be changed throughout the manuscript to ‘oxidation to carboxylic acid’. For instance, for table 1, is should be: ‘Table 1. Optimization of the oxidation reaction of compound 4 to carboxylic acid 5’.

  • We agree with your opinion, and it was corrected.

Page 5 – where is “The oxidation reaction of selenium dioxide” should be “The oxidation reaction with selenium dioxide”

  • We agree with your opinion, and it was corrected.

Page 5 – Please change “Table 4. Optimization of oxidation reaction conditions of compound 4” to “Table 4. Optimization of the reaction conditions for the oxidation of compound 4 to aldehyde 6”.

  • We agree with your opinion, and it was corrected.

Round 2

Reviewer 1 Report

The revised version of the manuscript is much better than the previous one. The paper is more complete and data are more clearly exposed. However, the opinion of this referee remains that the scientific content itself does not deserve publication in a journal with a IF close to 5. The paper reports just a synthesis optimization with known protocols (even if well adapted by authors to their scope) and thus the degree of novelty is not sufficient to fulfill with journal statement to "provide rapid publication of cutting-edge research". Although some minor concern points are still present (i.e. 1) the title has to be changed as "Suzuki-Miyaura Reaction" is only a minor point with the respect to the overall synthesis; 2) it sounds strange that 1,2-dimethoxyethane is "not suitable for large-scale synthesis" while in the following step pyridine is the selected solvent!), the paper in the present form can for sure find publication in a more suitable, less-impact and more specialized journal.

Author Response

  • Thank you for your comments.

    We revised our manuscript in accordance with your advices.

  •  

    We agree with you. However, because 1,2-dimethoxyethane is classified as a specific hazardous substance, it was changed to ethanol solvent. Since ethanol is harmless to the human body, it was used as a solvent in multi-kilogram production. In case of pyridine, it could be used in the next step because it is not a specific hazardous substance. However, care must be taken when using pyridine as it is flammable and irritating. And at this stage, it is possible to oxidize the methyl group to carboxylic acid in high yield using only SeO2/pyridine, and it has been developed as an industrial process capable of producing several kilograms.
  • We changed title to “Scalable Synthesis of TRPV1 Antagonist Bipyridinyl Benzimidazole Derivative via the Suzuki-Miyaura Reaction and Selective SeO2 oxidation”.
